# Training for Food Handlers at Production Level in Italian Regions

**DOI:** 10.3390/ijerph20032503

**Published:** 2023-01-31

**Authors:** Daniele Pattono, Matteo Petey, Anna Maria Covarino, Marta Gea, Tiziana Schilirò

**Affiliations:** 1Department of Veterinary Sciences, University of Torino, Largo Paolo Braccini 2, 10095 Grugliasco, TO, Italy; 2Prevention Department, Public Health Service Aosta, Località Amérique, 7/L, 11020 Quart, AO, Italy; 3Department of Public Health and Pediatrics, University of Torino, Piazza Polonia 94, 10126 Torino, TO, Italy

**Keywords:** training, food policy, food safety, food regulation, food handlers

## Abstract

Food safety has always been a public health challenge. Globally, food safety control is supported by laws and preventive measures, such as inspections conducted from primary production to market, “from farm to fork” as emphasized by the European Union and training of Food Handlers (FHs). This latter preventive measure plays a very important role, and for this reason a review of training courses regulations provided in the different Italian regions was conducted. Analysis of the results shows that the Italian regions approach this issue in different ways: some regions provide only general guidelines, while others offer detailed instructions. The most significant differences concern the topics dealt with, the stakeholders, the staff training and the verification of results; topics such as allergens and gluten are often absent. More detailed guidelines tailored to fit the local scenario could provide better support to FHs, thus leading to real changes in their behaviors and mindsets and promoting the development of an actual “prevention culture”.

## 1. Introduction

Foodborne diseases (FBDs) are reported every year [1] and remain a very important issue in the context of public health in both underdeveloped and industrialized countries. The World Health Organization (WHO) estimates that nearly one in ten people has an FBD and that five of the seven most recent public health emergencies of international concern (PHEIC) were caused by issues related to food. Contaminated food causes as many as 600 million foodborne illnesses, with an estimated global burden as high as 33 million disability-adjusted life years in 2010 [2]. With regard to a relevant number of outbreaks, restaurants and food production plants have been identified as the sources of the infections [3,4]. Many options have been considered to reduce the incidence of FBDs, starting with food safety.

Food safety refers to the handling, preparation and storage of food in a manner that minimizes the risk of people contracting FBDs, and it is ensured by the adoption of food safety practices. These practices have a positive impact on the economic growth of the area in which they are implemented; however, in order to be conceived and put into practice, they need not only sound science but also fair law enforcement. As technology advances, new rules must be implemented to ensure that the relevant laws remain consistent and that people have access to a safe and wholesome supply of food to maintain their health and well-being.

Since the 2000s, the European Union has focused on controlling the process of handling, preparation and storage of food rather than the product (i.e., the final food). For this reason, from the issuance of Reg 178/2002 [5] until the present, the adoption of the hazard analysis critical control point (HACCP) system has been emphasized in the food industry as a way of mitigating health risks [6]. In addition to the HACCP system, good manufacturing practices (GMP) and good handling practices (GHP) have gained importance in this field because human error has been proven to be a key factor associated with the incidence of FBDs [4,7].

In addition, to define processes and practices aimed at ensuring a safe food supply, to achieve good results in food safety and to bring about a real change in the context of public health and FBDs, it is necessary to provide training to food handlers (FHs) as well as education regarding food culture [8]. This activity must be addressed throughout the food preparation chain, considering all the stakeholders involved, from the managers to the personnel involved in the preparation, serving and cleaning operations. However, European regulations do not provide specific information about it. Therefore, in Italy at the present time, the training system regarding food safety is still under regional regulations [9,10,11,12,13,14,15,16,17,18,19,20,21,22,23,24,25,26,27,28,29,30] (i.e., Italian regions apply the general framework, which is set by general European mandatory rules, in different ways). Since specific guidelines would be useful to improve national prevention, the aim of this review is to provide an overview of the training system in Italy. In this regard, both national and regional legislation on food safety was taken into consideration, ranging from the production to the industrial distribution of food. The review also aims to critically analyze the different training systems adopted both in Europe and worldwide.

These revisions could be the basis for a careful reflection on all subjects involved in food safety, both in private (producers, operators, training agencies) and public (control bodies) companies.

## 2. Material and Methods

In October 2021, a systematic literature search of the titles, abstracts, keywords and full text (if available) of articles contained in five databases was conducted. These databases included PubMed (Medline), Cumulative Index of Nursing and Allied Health Literature (CINAHL), Abstracts of Food Science and Technology (FSTA), Food Science Resources (FSS) and Google Scholar. These databases were chosen because they cover various fields in the nutritional, public health and biomedical sciences. Terms used in the search algorithm included subject keywords identified by the PICO technique (*e*.*g*., “food safety”, “foodborne illness*”) and associated with relevant control methods (*e*.*g*., HACCP, “hazard analysis and critical control points*”, “food safety training*”, “food inspection*”). Reviews of European, national and regional databases were also conducted to retrieve information regarding plans and regulations at the European, national, regional and sub-regional levels (autonomous provinces). Open-access documents were obtained from the websites of the relevant institutions (*e*.*g*., European Community, national and regional official gazettes). For database analysis, the period under study was from 2011 to 2021. For each reference, several topics were analyzed for the purposes of comparison: the presence or absence of specific guidelines, the recipients of the training, the topics covered in the training, the presence or absence of guidelines regarding the training methods, the duration of training and frequency of updating, the providers of the training and the qualifying degrees necessary to obtain an exemption from training. Not all Italian regional plans or regulations cover each of the analyzed topics. For this reason, in the results and discussions sections, each topic is presented, comparing only the plans of the regions that covers that specific topic (excluding regions whose plans do not deal with the topic).

## 3. Results

FH training courses are one of the most reliable actions used to control FBDs and to create a long-lasting “food prevention culture”. The examination of regional regulations shows that the importance of this aspect seems to be partially neglected. Indeed, five Italian regions (Basilicata, Lombardy, Piedmont, Sardinia, Veneto and the autonomous province of Trentino) [9,10,11,12,13,14,15] provided specific guidelines, taking into account European and Italian laws but without specific indications at the regional level. The other Italian regions or autonomous provinces gave detailed instructions about FH training courses. These guidelines offered details regarding stakeholders, programs and contents, duration and frequency of courses, teaching staff, verification by public health services and specific exemptions.

### 3.1. Stakeholders of the Training

To define the relevant stakeholders, regions or autonomous provinces took into consideration the following factors: role in the plant, activity, risk assessment for FHs in the production plants, handling of foods in the plant and kind of food handled in the plant, and six different classes were identified:(1)FHs in food production plant without specifications (one region) [16];(2)FHs with a specific role in food production plant (*e*.*g*., pastry chef) (three regions) [17,18,19];(3)Risk assessment for FHs in the production plants (six regions and Bolzano autonomous province) [20,21,22,23,24,25,26];(4)Risk assessment for FHs in the production plants and FHs as defined by European Community laws (two regions) [27,28];(5)Risk assessment for FHs in the production plants and the FHs role in the food production plant (one region) [29];(6)Kind of plant and manager staff (*e*.*g*., production, selling, catering) (one region) [30].

The risk assessment level, high or low, for FHs of companies was based by public health service on type of products, range of commercialization, number of workers on the line, direct or indirect contact with food and FHs activity work.

A specific training course was stipulated by six regions for FHs working at temporary events, such as fairs, local festivals or cultural and sporting events, and only one region mandated a specific form of training for newly hired staff.

### 3.2. Topics Covered in the Training

Based on the classes discussed above, the regions implemented different programs, taking into account the risk assessment for FHs in the production plants. It is important to underscore that no Italian region provides indications or guidelines concerning training methods.

The topics of the training courses in each class and their frequencies in relation to the stakeholders of the training courses in the classes are shown in Table 1 (a few topics not present in all classes, *e*.*g*., architectural needs of food plants, were present in class n°1, so they have not been taken into account in the table).

Most frequently, topics were about practical tasks, such as FH hygiene, the environment or the handling of food and HACCP programs regardless of the trainee’s role in the plant (high-risk profile, low-risk profile or management). FBDs were emphasized less frequently, being included in only 50% of the programs regardless of the FHs role as well as the food legislation for which two regions emphasized this topic for FHs with high-risk profiles or managers (not for FHs with low-risk profiles). Notably, some topics, such as allergens or gluten, were neglected by many regions.

### 3.3. Duration of Training and Frequency of Refresher Courses

In general, the mean duration of training was 8 h; however, a wide range of durations could be observed. Only 1 region mandated 10 h of training for FHs (class N.1) without any other specification (handling, production, catering), which did not take into account the complexity of the activity. Other regions used different criteria to make decisions regarding the duration of training. The most frequently considered criterion pertaining to the duration of training was based on “risk assessment”: 10 regions ranging from 3 to 8 h (for FHs with low-risk profiles) and from 3 to 12 h (for FHs with high-risk profiles) and 3 regions (Abruzzo, Puglia and Sicily), ranging from 4 to 12 h (considering the frequency of the role of FHs in the food plants). Tuscany required training with durations ranging from 12 (low complexity) to 16 h (high complexity) as defined in Table 1, and the Friuli Venezia Giulia region considered whether training was delivered to catering and production FHs (8 h), to sales operators (4 h) or included a general assessment of “risk” (3 h).

The scheduled periods for refresher courses varied between 2 and 5 years depending on the risk profile (refresher courses for those with high-risk profiles were usually stipulated after a period of 3 years, although in the Liguria region, this period was every 5 years, and in the Friuli Venezia Giulia Region it was every 2 years). For three regions (Calabria, Emilia Romagna and Liguria), the refresher courses for low-risk operators were not considered at all.

### 3.4. Training Providers

Both private and public companies were considered to be training providers. In particular, 11 regions listed both types of companies, while 3 regions listed only private companies. Only one autonomous province offered any choice in this context.

The following public providers were listed: the public health service (10 regions) and in 1 region, the National Network of Public Health Laboratories (Istituto Zooprofilattico Sperimentale).

Private companies were classified into the categories of training institutions (12 regions), trade or professional associations (10 regions), internal quality assurance offices (8 regions) and private training companies (6 regions).

University degrees were required for training staff only in 7 of 15 regions. Each region considered more than 1 degree, but the number of degrees permitted in the regions ranged from 7 to 13 different sciences. Medical degrees (human medicine, veterinary science) or food sciences (food sciences and technologies, public health sciences and chemistry) were required in six regions. Other permitted degrees included pharmacy (five regions), biological science (five regions) and animal production or agronomy science (four regions). Some regions also accepted other degrees that were not strictly related to food safety or food production and instead focused more closely on food quality or food management, such as dietary science, food engineering or herbal techniques.

### 3.5. Qualifying Degrees Associated with Exemption from Training

Similar to degrees required for training staff, the degree necessary to be exempted from the training courses was also highly variable according to the different regional plans/regulations.

In particular, 9 of 15 regions defined the degrees that could grant the degree holder an exemption from training courses. Degrees in medicine were the most common members of this category, alongside degrees in agricultural science and biology (nine regions). Eight regions granted exemptions from training courses to graduates in prevention sciences or dietary techniques.

It is interesting that secondary school diplomas such as those granted by hotel and catering management schools or agricultural experts were also included in this category by seven and six regions, respectively.

### 3.6. Training Verification by a Competent Authority

The majority of regions employed a system of document control based on the verification of certificates (eight regions) or the HACCP manual or related material (four regions). Direct observation or interviews with FHs were reported by six and four regions, respectively. Six regions made no mention of such verification processes.

## 4. Discussion

The first aspect of food that must be guaranteed is food safety. This goal can be achieved through food safety regulatory systems and health protection measures [31]. Training is one of the most important ways to improve and obtain a long-lasting effect considering prevention of FBDs. At the moment, this activity is mandatory, but it is applied with several differences among Italian regions. Since general guidelines could be helpful to obtain a uniform application, we conducted a review of regulations in order to provide tools useful to design a comprehensive and common regulation on FH training.

The results of the present review showed that while some regions have implemented training without indicating in detail what is required by EU legislation, other regions have provided specifics regarding all or many of the points that can be taken into consideration in this context [32]. These differences among the different Italian regions can be explained considering that, as highlighted by other authors in other countries [8,31], the different Italian regions face a wide variety of situations. In order to better manage the different situations, in Italy a certain degree of autonomy is granted to different regions. This is an appropriate approach in large countries, while in contrast, in smaller countries, a low level of local autonomy is justified by the low level of diversity [8]. The choice between a high and low degree of autonomy is not merely a question of extension but also a matter of budget [31]. Indeed, a more detailed system can be costlier than a less detailed system [8].

In general, the Italian training programs focus mainly on practical issues that are faced every day by workers. The subjects included in these programs are in line with what has been reported in the literature regarding other countries. GHP, GMP, cleaning and disinfection procedures and HACCP plans have been identified as fundamental for imparting knowledge and raising the level of food safety in production at different stages of the food production process [33]. These subjects are considered to be relevant for both profiles (high- and low-risk workers and plants), as noted by De Andrade et al. [34]. Legislation, HACCP and FBDs are considered to be less important by half of the regions, in particular for low-risk FHs; however, they are viewed as very important for FHs with high-risk profiles and management. Instead, these topics are included in the training programs offered in other nations [35]. Notably, some topics, such as allergens and gluten, are not considered at all or are considered only in a few cases or by a few regional training programs. In our opinion, these subjects are relevant, considering the problems resulting from the cross-contamination of lines, which are becoming increasingly important [36,37]. The same consideration applies to traceability. Considering the fact that recalls are one of the most important measures used by plants to minimize food hazards, in our opinion, this point should also be included, especially for lower-level FHs who are directly involved with this procedure [31]. However, training programs must also be chosen on the basis of the relevant territory and the local situation [8,31]. Some authors emphasize the importance of an analysis of specific needs to the production of a more reliable assessment when choosing the subjects of training courses [33].

Regarding the choice of participants, the inclusion of food operation managers and supervisors in addition to FHs (*e*.*g*., in the Campania region) is a policy that is followed and recommended in cases in which the training must be adapted to particular subjects and functions [3,34,38]. The exclusion of people with high school degrees or university degrees as managers seems not to be a win–win policy because the inclusion of people with high school degrees has a positive impact on the level of the food safety culture exhibited by all participants [34,35].

Choosing different teaching strategies (*e*.*g*., only theorical lessons, group work, problem solving, practical cases analysis) based on behavioral theories can be a useful way of achieving positive results. Our analysis shows that this aspect is not taken into account in regional guidelines. Research has emphasized the fact that the results do not always involve a change in behavior either in whole or in part [34].

In particular, simple activities (*e*.*g*., handwashing) achieve higher success rates than more complex activities [3].

At present, three main behavioral theories can be recognized:(1)The health belief model, according to which an individual performs a preventive behavior based on their desire to avoid illness (or to recover from illness) and their belief that a specific health action can prevent (or ameliorate) illness.(2)The knowledge, attitude and practice (KAP) model, which assumes that an individual’s behavior or practice is dependent on their knowledge (K) and suggests that the mere provision of information leads directly to a change in attitude (A) and, consequently, a change in behavior or practice (P).(3)The theory of planned behavior (TPB), which focuses on the individual’s intention to perform a given behavior and has been used by many researchers to predict the determinants of an FH’s behavior [39].

Among these three models, KAP seems to exhibit a variety of important biases [40]. Several authors have noted that whichever model is adopted, it is important for training interventions to be able to enhance risk perceptions [39]. For example, successful training intervention should be designed combining theory and practical activities or combining multiple channels and methods in order to ensure that the limitations of a single method are eliminated or at least reduced [6,40]. Additionally, a teaching strategy that includes mediation and moderation effects due to the addition of a third variable, such as experience, job satisfaction or motivation, is able to facilitate the relationship between knowledge and application [40]. In addition, to develop a win–win strategy, different actors must share a common language, and the results must be checked using public health impact assessment to facilitate the coordination of policies related to public health outcomes [38].

Training could be assessed in different ways, and according to our review, this aspect is not taken into account by many regions (6 of 15 regions, i.e., 40% of the total). Among regions that took this aspect into account, the selected methods were documental control of training certificates associated with HACCP manuals, interviews or the direct observation and application of GHP. The first method has been reported to be less efficient because it is not possible to observe real behavioral changes using this approach. Additionally, interviews as compliance checks are not useful due to the fact that self-reporting practices are not a reliable source of data and are frequently associated with no improvements in observed behaviors [3,40]. Very few regions, i.e., only six, used direct observation of compliance with GHP or laboratory tests (*e*.*g*., microbiological analysis in the case of handwashing practices), which have been reported to be more reliable [6,35,40]. In our opinion, this strategy must be taken into account in guidelines for training courses.

Our final consideration pertains to the validity of training courses and their repetition. This aspect was emphasized by all regions that considered the risk profiles of FHs, which mandated new training sessions after a period of 2–5 years. In our opinion, this is a wide range that could create discrepancies among the regions and/or autonomous provinces. Regarding the timing and duration of training as well as the scheduled date for the repetition of training, our results show that these times are longer than those reported in the literature, which indicated the mean time for catering to be 3 h with a suggested time of 4 h per lesson and refresher courses scheduled after a period ranging from 6 months to 1 year [34]. A short time between training sessions is necessary since some behavioral changes can disappear a few months after training [6]. Several authors have noted that repeated training sessions or new sessions can reinforce behavioral changes and improve the enforcement of food safety policy [39]. For these reasons, this aspect must also be taken into account when developing relevant guidelines.

## 5. Conclusions

Food safety is an important global challenge for public health. The control of food safety requires the integration of many public and private systems at several levels, which must collaborate to ensure ultimate success [31]. All these systems are based on both objective and subjective aspects such as national laws, inspection systems and training. To achieve results that are more positive every year, governments need to implement year-by-year control measures aimed at prevention and mitigation.

Among these measures, training is probably the only intervention that can ensure real change by developing a “culture of prevention”; however, it is necessary to involve FHs in this approach by motivating them and increasing their personal dedication to the culture of prevention [34]. FHs must be instilled with common purposes and provided with systems for controlling the results while simultaneously taking local conditions into account. The relevant training must include specific topics, motivated teaching staff and audiences and a good methodology [8]. Our analysis provides evidence of several differences through the Italian territory.

According to the different Italian regions, we believe that various aspects of training courses for FHs that are seldom considered should be taken into consideration; for example they should include topics such as food intolerance and allergies, and as highlighted by numerous authors, they should also include training for managers. The duration and cadence of the courses and refresher courses should be considered in a clear manner, considering commercial exchanges in order to obtain uniformity. If these recommendations will be implemented in training courses, the behavioral changes arising from them may not only be a change in the food production and handling process, but also a change in the mentality of the operators and therefore a real and lasting change towards an effective “culture of prevention”.

## Figures and Tables

**Table 1 ijerph-20-02503-t001:** Frequency of topic coverage by training courses for each profile (green: 100% to 76%, yellow: between 75% and 51%, orange: between 50% and 26%, red: between 25% and 0%).

		Topics										
Classes	Stakeholders	GMP	FHs Hygiene	Environmental Hygiene	HACCP	Food Hazards	FBDs	Traceability	Gluten	Allergens	Food Laws	FH Duties
1 (1 Region)		100%	100%	0%	100%	100%	100%	0%	0%	0%	100%	0%
2 (3 Regions)	FHs	100%	100%	100%	66%	66%	66%	33%	33%	33%	33%	33%
3 (6 Regions and Bolzano autonomous province)	High-risk plant FHs	85%	85%	85%	85%	71%	42%	42%	28%	14%	57%	42%
	Low-risk plant FHs	85%	85%	71%	42%	57%	42%	14%	28%	0%	71%	42%
4 (2 Regions)	High-risk plant FHs	100%	100%	100%	100%	0%	50%	0%	0%	0%	100%	50%
	Low-risk plant FHs	100%	50%	100%	100%	0%	50%	0%	0%	0%	50%	50%
	FHs as defined by EU Reg*	100%	100%	100%	100%	50%	50%	0%	0%	0%	100%	50%
5 (1 Region)	High risk plant FHs	100%	100%	100%	100%	100%	0%	0%	0%	0%	0%	0%
	Low-risk plant FHs	100%	100%	100%	100%	100%	0%	0%	0%	0%	0%	0%
	Simple activity FHs	100%	100%	100%	100%	100%	0%	0%	0%	0%	0%	0%
	Complex activity FHs	100%	100%	100%	100%	100%	0%	0%	0%	0%	0%	0%
6 (1 Region)	High-risk plant FHs	100%	100%	0%	0%	0%	100%	0%	0%	0%	0%	0%
	Manager	100%	0%	0%	100%	100%	0%	0%	0%	0%	0%	100%

Food laws, “Hygiene Package” as defined by EU Reg 178/2002 and others; FHs, Food Handlers; GMP, good manufacturing practices; FBDs, foodborne diseases; simple activity, *e*.*g*., sale, distribution and storage of food; complex activity, *e*.*g*., production and preparation of food.

## Data Availability

Our database is reported in the References Section (national and regional laws database).

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
