# Peer review of "Training for Food Handlers at Production Level in Italian Regions"

_ijerph, 2023, doi:10.3390/ijerph20032503_

Round 1

Reviewer 1 Report

According to the title, the main question is the training of food handlers in Italy. I assume from the title that authors get information about types, duration, standards or anything related to the food safety held in Italy. But, the materials I found in materials and methods and results are like a review paper (got all information from various search engines).  I was assuming that It will be a survey about food safety by food handlers. Author should consider this data as a review article not survey. References are fine.

Its very confusing. Title seems to provide information about training for food handlers in Italy. It seems for title that authors will provide information about type of training, their durations, person involved, standard of different trainings already organized for various food handlers related to persons involved in harvesting, in distribution, in marketing, in processing and in sale. But all the data provide is searched from various search engines. 

How this data covers the various regions of Italy?

Author Response

Dear editor here enclosed there are the answers to Reviewer n. 1

Thank you very much.

Reviewer 2 Report

Dear Sir,

From the beginning of accepting to review ijerph-2113748, I have spent 8 days to persuade myself to seek the highlights of this article, But after reading the 《GUIDELINES: ETHICAL GUIDELINES FOR PEER REVIEWERS》of MDPI AND IJERPH, In the end, I believe that rejecting articles is the real responsible behavior for periodicals, authors and readers, for the following reasons:

1. After focusing on the conclusion in the abstract and the conclusion at the end of the paper, I personally believe that the research value is very limited;

2. The research methods are not introduced clearly, which cannot prove that the proposed views are correct;

3. Methods, experiments and conclusions cannot be reliable, comprehensive and reasonable;

4. The research is not supported by enough appropriate theories and literature;

Author Response

Dear editor here enclosed you find the answers to reviewer n. 2.

Thank you very much.

Reviewer 3 Report

Comments to Authors:

This article states the importance and quality and also the methods of training to food handlers in relation to food safety in Italy. Through training, food handlers gain important knowledge and skills that protect the public, themselves and their families from illness.

 The manuscript is suitable for publication in International Journal of Environmental Research and Public Health.

Some recommendations which can be addressed are listed below:
1- H
ighlight the novelty of the manuscript in the text. 

2- Define food handlers in introduction, for example: Food handlers refers to anyone who is involved in any activity that involves food or surfaces likely to come in contact with food, including those in:

  • preparing – such as chopping, cooking, thawing
  • delivering and transporting
  • packing
  • serving
  • cleaning tableware or equipment that comes in contact with food.

Or anyone who, through their work activity, has direct contact with food during any of its phases until it reaches the final consumer. This includes: preparation, manufacture, processing, packaging, storage, transport, distribution, sale, supply and service.

Author Response

Dear editor

here enclosed we uploaded the answers to reviewer n. 3

Thank you very much.

Round 2

Reviewer 2 Report

      It seems that the author has expertly redacted inappropriate details in the original text, so I suggest to has agreed to accept it for publication